# Fault Diagnosis of Planetary Gearbox Based on Hierarchical Refined Composite Multiscale Fuzzy Entropy and Optimized LSSVM

**DOI:** 10.3390/e27050512

**Published:** 2025-05-10

**Authors:** Xin Xia, Xiaolu Wang

**Affiliations:** 1School of Mechanical and Electrical Engineering, Suqian University, Suqian 223800, China; 2Information Construction Center, Suqian University, Suqian 223800, China; wangxiaolu@squ.edu.cn

**Keywords:** hierarchical entropy, refined composite multiscale fuzzy entropy, optimized LSSVM, fault diagnosis, planetary gearbox

## Abstract

Efficient extraction and classification of fault features remain critical challenges in planetary gearbox fault diagnosis. A fault diagnosis framework is proposed that integrates hierarchical refined composite multiscale fuzzy entropy (HRCMFE) for feature extraction and a gray wolf optimization (GWO)-optimized least squares support vector machine (LSSVM) for classification. Firstly, the HRCMFE is developed for feature extraction, which combines the segmentation advantage of hierarchical entropy (HE) and the computational stability advantage of refined composite multiscale fuzzy entropy (RCMFE). Secondly, the hyperparameters of LSSVM are optimized by GWO using a proposed fitness function. Finally, fault diagnosis of the planetary gearbox is achieved by the optimized LSSVM using the HRCMFE-extracted features. Simulation and experimental study results indicate that the proposed method demonstrates superior effectiveness in both feature discriminability and diagnosis accuracy.

## 1. Introduction

Planetary gearboxes are deployed extensively across multiple industrial sectors, such as wind turbines, marine vessels, and aerospace systems. In real world applications, faults in planetary gearboxes occur frequently [1]. If these faults remain undetected and unaddressed, they can potentially escalate into large scale accidents [2]. Thus, research on the fault diagnosis of planetary gearboxes is of substantial importance.

Rotating mechanical systems emit vibration signals during operation, which encode comprehensive information about their operational status [3]. The majority of contemporary fault diagnosis studies for planetary gearboxes focus on analyzing vibration signals [4,5,6]. Planetary gearboxes utilize multiple identical planet gears orbiting a central sun gear, while simultaneously engaging with multiple gear pairs on the planet carrier. Owing to their intricate transmission architectures and dynamic operating conditions, the vibration signals of planetary gearboxes demonstrate non-linear and non-stationary properties [7]. Fault diagnosis techniques for planetary gearboxes utilizing vibration signals primarily encompass characteristic frequency-based methods [8,9,10,11,12], deep learning-driven approaches [13,14], and entropy-based methodologies [15,16]. Characteristic frequency-based methods are distinguished by their explicit physical interpretability and robust diagnostic efficacy for typical faults. Nevertheless, their performance heavily depends on precise theoretical models and accurate operating condition parameters which limit their adaptability to scenarios with uncertain or variable operational contexts. In contrast, deep learning-based fault diagnosis frameworks eliminate the reliance on mechanical models and parameterization, offering data-driven autonomy. However, these methods demand substantial labeled datasets for training, posing significant challenges to practical implementation, especially in industrial settings where large-scale annotated data acquisition is often costly, time-consuming, or even infeasible due to safety or accessibility constraints.

Entropy-based non-linear dynamic methods have demonstrated efficacy in feature extraction for fault diagnosis [15]. Many researchers have applied entropy-based feature extraction to planetary gearbox fault diagnosis [17,18,19,20]. Sample entropy (SE) is characterized by high computational efficiency and robust noise resistance [21]. However, it is prone to substantial bias when applied to short time-series datasets [22]. Permutation entropy (PE) exhibits sensitivity to dynamic changes in signals but neglects amplitude information, which may result in partial loss of discriminative details [23]. Moreover, PE imposes significant computational complexity for long sequences [24]. Dispersion entropy (DE) demonstrates efficient computation and noise robustness but encounters challenges related to high computational complexity and parameter selection for categorical partitioning [25]. Fuzzy entropy (FE) effectively addresses data uncertainty and exhibits robust noise resistance, making it well-suited for high-noise operational environments [26]. However, single-scale entropy metrics are often insufficient to comprehensively capture the information encoded within the complex vibration signals [27]. To address this limitation, time-series-based multiscale entropy approaches have been developed to augment feature extraction capabilities [28,29,30,31]. Furthermore, frequency-domain-based multiscale entropy frameworks such as hierarchical entropy (HE) have also emerged [32]. During multiscale computation, time-series coarse-graining reduces data length, which can lead to instability in entropy calculations. To address this limitation, refined composite multiscale entropy (RCME) has been proposed to enhance the stability of entropy estimation [33,34,35]. By combining the segmentation advantage of hierarchical entropy in frequency domains with the computational stability advantage of multiscale refined composite fuzzy entropy, a hierarchical refined composite multiscale fuzzy entropy (HRCMFE) was proposed in this study, which can extract more useful fault information from planetary gearboxes.

Many machine learning methods have been applied to automate the classification of operational states of planetary gearboxes [36,37,38]. The artificial neural networks (ANN) exhibit strong non-linear fitting capabilities in handling complex classification problems. However, they impose significant demands for extensive training datasets, entail intricate training procedures, and are susceptible to overfitting [39]. The K-Nearest Neighbor (K-NN) algorithm dispenses with explicit data training, a notable advantage in scenarios with limited labeled data [40]. Nonetheless, it entails computationally intensive pairwise distance computations between all test samples and training instances [41]. Fault training samples for rotating machinery, such as planetary gearboxes, are often scarce, making classifiers suitable for small sample sizes attract extensive attention from researchers [42,43]. The Support Vector Machine (SVM) excels in addressing non-linear problems and demonstrates effectiveness in classifying high-dimensional and small-sample datasets. However, due to its reliance on solving quadratic programming problems during training, SVM incurs significant computational complexity [44]. The least squares support vector machine (LSSVM) can significantly reduce computational complexity and improve numerical stability compared to SVM [45]. However, the classification performance of LSSVM is susceptible to the hyperparameters setting [46]. Some researchers have focused on developing advanced hyperparameters optimization strategies for LSSVM to enhance its classification capabilities [47,48].

In this paper, a fault diagnosis framework is proposed that integrates HRCMFE for feature extraction and a gray wolf optimization (GWO)-optimized LSSVM for classification. Firstly, the HRCMFE is proposed for feature extraction. Secondly, the hyperparameters of LSSVM are optimized by GWO using a proposed fitness function. Finally, fault diagnosis of the planetary gearbox is achieved by the proposed fault diagnosis method. The proposed method demonstrates superior effectiveness in feature extraction and fault classification.

The paper is structured as follows: related theory and the proposed HRCMFE are introduced in Section 2. In Section 3, the GWO-LSSVM and the fault diagnosis framework are presented. Simulation studies are presented to verify the effectiveness of the proposed HRCMFE in Section 4. In Section 5, fault diagnosis by the proposed method is analyzed and discussed. Finally, the conclusion and main contributions are drawn in Section 6.

## 2. Hierarchical Refined Composite Multiscale Fuzzy Entropy

### 2.1. Multiscale Fuzzy Entropy

Fuzzy entropy is a powerful approach to evaluate the complexity of time series, distinguished by its remarkable computational stability. Multiscale fuzzy entropy, integrating the concepts of fuzzy entropy and multiscale analysis, delves into the dynamic properties of signals across diverse time scales [34]. This integration enables a more comprehensive and in-depth characterization of the rich tapestry of characteristic information inherent in real-world signals, thereby providing a more complete understanding of their underlying dynamics. Assume a time series $u=\left\{u_{1},u_{2},\ldots ,u_{L}\right\}$ with *L* data length; the multiscale fuzzy entropy (MFE) of time series *u* can be calculated as follows:

(1) New data series $x^{\left(\tau \right)}$,with scale factor *τ*, are obtained by the coarse-graining process, as follows:(1)$$x_{i}^{\left(\tau \right)}=\frac{1}{\tau }\sum_{b=(i-1)\tau +1}^{i\tau }u_{b},\;\;\;\;\;\;\;\;\;\;\;where\;\;\;\;\;i=1,2,\ldots ,(N=\left\lfloor \frac{L}{\tau }\right\rfloor )$$
where *N* is the length of series $x^{\left(\tau \right)}$.

(2) The data series $x^{\tau }$ are mapped into high-dimensional space according to Takens embedding theorem, as follows:(2)$$\left\{\begin{array}{l} S_{i}^{m}=[x_{i},x_{i+1},\ldots ,x_{i+m-1}]-{\bar{x}}_{i},\;\;\;\;\;\;\;\;\;\;i=1,2,\ldots ,N-m+1\\ {\bar{x}}_{i}=\frac{1}{m}\sum_{j=0}^{m-1}x_{i+j} \end{array}\right.$$
where *m* represents the embedding dimension, and it usually takes values between 2 and 5.

(3) The similarity distance can be calculated by the Chebyshev distance function, as follows:(3)$$\begin{array}{l} d_{i,j}^{m}[S_{i}^{m},S_{j}^{m}]=\underset{p=0,1,\ldots ,N-m}{\max}\left[\left|S_{i}^{m}(i+p)-S_{j}^{m}(j+p)\right|\right]\\ where\;\;\;\;i,j=1,2,\ldots ,N-m+1\;\;\;and\;\;\;i\neq j \end{array}$$
where $d_{i,j}^{m}[S_{i}^{m},S_{j}^{m}]$ is the similarity distance between $S_{i}^{m}$ and $S_{j}^{m}$.

(4) The fuzzy similarity distance can be calculated according to the fuzzy function as follows:(4)$$A_{i,j}^{m}(d_{i,j}^{m})=\exp\left[-{\left(\frac{d_{i,j}^{m}}{r}\right)}^{n}\right]$$
where *n* and *r* represent the boundary gradient and similarity tolerance, respectively.

(5) The similarity degree is calculated as follows:(5)$$\phi^{m}(n,r)=\frac{1}{N-m}\sum_{i=1}^{N-m}\left(\frac{1}{N-m-1}\sum_{j=1,j\neq i}^{N-m}A_{i,j}^{m}\right)$$

(6) The similarity degree for embedding dimension *m* + 1 is calculated following the same procedure, and the fuzzy entropy and multiscale fuzzy entropy can be defined as follows:(6)$$\left\{\begin{array}{l} MFE(u,m,n,r,\tau )=FuzzyEn(m,n,r,x^{\tau })\\ FuzzyEn(m,n,r,x^{\tau })=-\ln\left(\frac{\phi^{m+1}(n,r)}{\phi^{m}(n,r)}\right) \end{array}\right.$$

### 2.2. Hierarchical Multiscale Fuzzy Entropy

The MFE uses the mean value to represent each signal segment during the coarse-graining process. This approach only accounts for the low-frequency components, failing to sufficiently consider the high-frequency components. As a result, it cannot effectively utilize the characteristic features of the original signal. Hierarchical entropy can simultaneously consider both the high-frequency and low-frequency components of the signal, enabling it to provide more comprehensive information about the signal patterns [32]. Therefore, by combining the hierarchical segmentation advantage of hierarchical entropy with the computational stability of MFE, hierarchical multiscale fuzzy entropy (HMFE) is proposed. Assume a time series $y=\left\{y_{1},y_{2},\ldots ,y_{M}\right\}$ with data length *M* = 2*^k^* (where *k* is a positive integer), the HMFE of time series *y* can be calculated by the following steps:

(1) The averaging operators are defined as follows:(7)$$\left\{\begin{array}{l} Q_{0}(y)=\frac{y(2j)+y(2j+1)}{2},\;\;\;\;\;j=0,1,2,\ldots ,2^{k-1}\\ Q_{1}(y)=\frac{y(2j)-y(2j+1)}{2},\;\;\;\;\;j=0,1,2,\ldots ,2^{k-1} \end{array}\right.$$

The length of $Q_{0}$ and $Q_{1}$ is $2^{k-1}$. The original data series can be reconstructed from $Q_{0}$ and $Q_{1}$ as follows:(8)$$y=[(Q_{0}{\left(y\right)}_{j}+Q_{1}{\left(y\right)}_{j}),(Q_{0}{\left(y\right)}_{j}-Q_{1}{\left(y\right)}_{j})],\;\;\;\;\;\;\;\;\;j=0,1,2,\ldots ,2^{k-1}$$

When *j* = 0 or *j* = 1, the operator *Q* has the following matrix representation:(9)$$Q_{j}(y)={\left[\begin{array}{ccccccc} \frac{1}{2} & \frac{{\left(-1\right)}^{j}}{2} & 0 & 0 & \ldots & 0 & 0\\ 0 & 0 & \frac{1}{2} & \frac{{\left(-1\right)}^{j}}{2} & \ldots & 0 & 0\\ \vdots & \vdots & \vdots & \vdots & \ddots & \vdots & \vdots \\ 0 & 0 & 0 & 0 & \ldots & \frac{1}{2} & \frac{{\left(-1\right)}^{j}}{2} \end{array}\right]}_{2^{k-1}\times 2^{k}}$$

(2) A *h*-dimensional vector is constructed as $[\gamma_{1},\gamma_{1},\ldots ,\gamma_{h}]\in \left\{0,1\right\}$. The nonnegative integer can be described as:(10)$$e=\sum_{j=1}^{h}\gamma_{j}2^{h-j}$$
where *h* represents the number of hierarchies. Equation (10) also indicates a nonnegative integer *e* which corresponds to an *h*-dimensional vector.

(3) For a given $h\in N^{+}$ and a nonnegative integer *e*, the original time series can be defined by hierarchical components as follows:(11)$$y_{h,e}=Q_{\gamma_{h}}•Q_{\gamma_{h-1}}•\cdots •Q_{\gamma_{1}}(y)$$
where $y_{h,e}$ represents the hierarchical components of the *h*-th layer. The schematic diagram of hierarchical segmentation is shown in Figure 1.

(4) The MFE of each hierarchical component is calculated to form the hierarchical multiscale fuzzy entropy, as follows:(12)$$HMFE(y,m,n,r,\tau )=MFE(y_{h,e},m,n,r,\tau )$$

### 2.3. Hierarchical Refined Composite Multiscale Fuzzy Entropy

When hierarchical segmentation is performed, the relatively long original signal is divided into shorter segments which will lead to significant fluctuations in the entropy value at relatively large scale factors. Therefore, the calculation framework of refined composite multiscale fuzzy entropy can be adopted to address the problem of entropy value fluctuations. The calculation steps of HRCMFE are as follows:

(1) Assuming a hierarchical components series $w=\left\{w_{1},w_{2},\ldots ,w_{L}\right\}$, the scale factor is *τ*. The *τ* coarse-grained sequences are obtained as follows:(13)$$\left\{\begin{array}{l} x_{a,i}^{\tau }=\frac{1}{\tau }\sum_{k=(i-1)\tau +a}^{i\tau +a-1}w_{k}\\ where\;\;\;\;\;\;a=1,2,\ldots ,\tau \;\;\;\;\;\;and\;\;\;\;\;\;\;i=1,2,\ldots ,(N=\left\lfloor \frac{L}{\tau }\right\rfloor ) \end{array}\right.\;\;\;\;\;$$

(2) The average values of ${\bar{\phi }}^{m}$ and ${\bar{\phi }}^{m+1}$ under different *τ* are calculated according to Equations (3)–(5). Finally, the HRCMFE is calculated as follows:(14)$$\left\{\begin{array}{l} RCMFE(w,m,n,r,\tau )=-\ln\left(\frac{{\bar{\phi }}^{m+1}}{{\bar{\phi }}^{m}}\right)\\ HRCMFE(y,m,n,r,\tau )=RCMFE(y_{h,e},m,n,r,\tau ) \end{array}\right.$$

## 3. The Proposed Fault Diagnosis Method

### 3.1. LSSVM

LSSVM is an advanced modification derived from the SVM, which significantly enhances computational efficiency and convergence accuracy while maintaining robust non-linear fitting capability and strong robustness [45]. Assume that there exists a training set $Sample=\{(x_{i},y_{i})\;\;\;\left|\;\;\;i=1\right.,2,\ldots ,N\}$, where $x_{i}$ is the input value and $y_{i}$ is the output value. The detailed procedures of LSSVM are as follows:

(1) A linear regression function is employed to map the input data to a high-dimensional feature space. The linear regression function is described as follows:(15)$$y(x)=\omega^{T}\varphi (x)+b$$
where $\omega^{T}$ is the weight vector, *φ*(·) is a non-linear mapping function, and *b* represents the bias term.

(2) The fitting optimization problem can be formulated as a least squares regression problem, as follows:(16)$$\left\{\begin{array}{l} \min J(\omega ,e)=\frac{1}{2}\left\|\omega \right\|+\frac{\gamma }{2}\sum_{i=1}^{N}e_{i}^{2}\\ s.t.\;\;\;\;\;y_{i}=\omega^{T}\varphi (x_{i})+b+e_{i},\;\;\;\;\;i=1,2,\ldots ,N \end{array}\right.$$
where γ represents the regularization parameter, and $e_{i}$ is error.

The Lagrangian form of Equation (16) can be described as:(17)$$L(\omega ,b,e_{i},\alpha_{i})=\frac{1}{2}\left\|\omega \right\|+\frac{\gamma }{2}\sum_{i=1}^{N}e_{i}^{2}-\sum_{i=1}^{N}\alpha_{i}\left[\omega^{T}\varphi (x_{i})+b+e_{i}-y_{i}\right]$$
where $\alpha_{i}$ represents Lagrange multipliers.

(3) According to Karush–Kuhn–Tucker conditions, the optimal solutions can be described as:(18)$$\left\{\begin{array}{l} \frac{\partial L}{\partial \omega }=0\to \omega =\sum_{i=1}^{N}\alpha_{i}\varphi (x_{i})\\ \frac{\partial L}{\partial b}=0\to \sum_{i=1}^{N}\alpha_{i}=0\\ \frac{\partial L}{\partial e_{i}}=0\to \alpha_{i}=\gamma e_{i}\\ \frac{\partial L}{\partial \alpha_{i}}=0\to \omega^{T}\varphi (x_{i})+b+e_{i}-y_{i}=0 \end{array}\right.$$

(4) Then, the equation of the LSSVM multivariate non-linear regression function can be described as follows:(19)$$f(x)=\sum_{i=1}^{N}\alpha_{i}K(x_{i},x,\sigma )+b$$
where *K*(·) is the kernel function, and σ is the bandwidth parameter of the kernel function.

As described above, the regularization parameter γ and bandwidth parameter σ will affect the classification ability of LSSVM. Therefore, it is necessary to optimize the parameters of LSSVM.

### 3.2. GWO-LSSVM

In this study, gray wolf optimization (GWO) was applied to optimize the hyperparameters of LSSVM.

The GWO algorithm, proposed by Mirjalili et al., is a novel bio-inspired heuristic algorithm derived from the predatory behavior of gray wolf populations [49]. Based on the hierarchical leadership structure observed in wolf packs, three dominant individuals (*α*, *β*, *δ*) and subordinate members (*ω*) are defined. The α wolf represents the global optimal solution, while *β* and *δ* wolves serve as suboptimal solutions. The remaining *ω* wolves act as candidate solutions. The algorithm initiates the search process by guiding the *ω* wolves toward the prey (optimal solution) in the search space, with the *α*, *β*, and *δ* wolves initially serving as the initial solutions for guidance. Through iterative position updates and continuous movement, the wolf pack progressively converges toward the global optimal solution.

The distance between the prey and the gray wolves can be described as follows:(20)$$\left\{\begin{array}{l} D=\left|C\cdot X_{p}(t)-X(t)\right|\\ X(t+1)=X_{p}(t)-A\cdot D \end{array}\right.$$
where *D* represents the distance between the gray wolf and prey, *t* represents the number of iterations, $X_{p}(t)$ represents the current position vector of the prey after *t* iterations, and X(t) is the current position vector of the gray wolf after *t* iterations. **A** and **C** are coefficient vectors subject to the following equations:(21)$$\left\{\begin{array}{l} A=2\cdot a\cdot r_{1}-a\\ C=2\cdot r_{2} \end{array}\right.$$
where *a* is a control parameter that linearly decreases from 2 to 0 during the iterative process. $r_{1}$ and $r_{2}$ are random vectors taking values in [0, 1].

The *α*, *β*, and *δ* wolves have the best solutions in each iteration, and will lead the rest of the wolves to update their positions, as follows:(22)$$\left\{\begin{array}{l} D_{\alpha }=\left|C_{1}\cdot X_{\alpha }-X(t)\right|\\ D_{\beta }=\left|C_{2}\cdot X_{\beta }-X(t)\right|\\ D_{\delta }=\left|C_{3}\cdot X_{\delta }-X(t)\right|\\ X_{1}=X_{\alpha }-A_{1}\cdot D_{\alpha }\\ X_{2}=X_{\beta }-A_{2}\cdot D_{\beta }\\ X_{3}=X_{\delta }-A_{3}\cdot D_{\delta }\\ X(t+1)=\frac{X_{1}+X_{2}+X_{3}}{3} \end{array}\right.$$
where $X_{\alpha }$, $X_{\beta }$, and $X_{\delta }$ are the positions of the *α*, *β*, and *δ* wolves, respectively. $D_{\alpha }$, $D_{\beta }$, and $D_{\delta }$ are the distances between the current wolf and the *α*, *β*, and *δ* wolves, respectively. X(t+1) represents the position of the gray wolf after the (*t* + 1)-th iteration.

The flowchart of the proposed GWO-LSSVM algorithm for parameter optimization is shown in Figure 2.

The detailed steps of GWO-LSSVM are as follows:

Step 1: The size of the wolf population, the maximum number of iterations *maxit*, and the range of the optimized parameters are initialized.

Step 2: The fitness values of individual gray wolves are calculated based on the fitness function and cross-validation is employed to calculate the fitness values to enhance the generalization ability of the LSSVM model.

The training feature and label datasets are divided into *M* subsets. Each subset is used in turn as the test dataset, while the remaining subsets serve as the training datasets for LSSVM training. The misdiagnosis rate of each subset test is evaluated, and its average value is calculated to serve as the fitness function value. The steps can be described as follows:(23)$$\left\{\begin{array}{l} f(\gamma ,\sigma ,M)=\frac{1}{M}\sum_{i=1}^{M}\varepsilon_{i}\\ \varepsilon_{i}=Lssvm\_model(subset_{i})\\ Lssvm\_model=train(subset_{j},\gamma ,\sigma ),\;\;\;\;\;\;\;j=1,2,\ldots ,M\;\;\;\;and\;\;\;\;\;j\neq i \end{array}\right.$$
where f(γ,σ,M) is the fitness value, $\varepsilon_{i}$ is the test error rate of the *i*-th subset $subset_{i}$, and Lssvm_model represents the trained model with parameters γ and σ.

Step 3: Gray wolves are ranked according to their fitness values, and the top three individuals with the highest fitness are selected and designated as *α*, *β*, and *δ* wolves in descending order of fitness.

Step 4: If the maximum iteration number has not been reached, the positions of the wolf population and adjustable parameters *a*, **A**, and **C** are updated, and then the algorithm returns to Step 2.

Step 5: If the maximum iteration number is reached, the parameter values of γ and σ are output according to the position of the *α* wolf.

### 3.3. The Proposed Fault Diagnosis Framework

In this paper, a fault diagnosis method based on HRCMFE and GWO-LSSVM is proposed for planetary gearboxes; the detail framework of the proposed method is shown in Figure 3.

The specific implementation steps include:(1)The proposed HRCMFE is employed to extract fault features from vibration signals of planetary gearboxes, which can effectively discriminate between high-frequency and low-frequency signal characteristics. Meanwhile, the refined composite computational framework significantly enhances the computational stability of entropy values.(2)GWO is employed to optimize the hyperparameters of LSSVM, and the fitness function of GWO is based on cross-validation. The proposed GWO-LSSVM can significantly improve the classification accuracy and generalization ability of LSSVM.(3)The proposed method is verified by the vibration signal of planetary gearbox under different states.

## 4. Simulation Study

In this section, the entropy stability of the proposed HRCMFE is verified through simulation by comparing it with HMFE and hierarchical multiscale sample entropy (HMSE). The software for RCMFE, MSE, and MFE can be obtained from the following link: https://github.com/HamedAzami/NLDyn/blob/main/Final_V10p.rar (accessed on 8 April 2025). The software for HRCMFE, HMSE, and HMFE were encoded in this study based on RCMFE, MSE, and MFE. The mean value and standard deviation (SD) of different entropies for 30 groups of white noise are calculated. The parameters of different methods are listed in Table 1, and the results for different data lengths are shown in Figure 4.

In the coarse-graining process of white noise multiscale entropy, mean values are employed, leading to entropy values that should decrease smoothly with increasing scale. As illustrated in Figure 4, the mean entropy curves of HMFE and HRCMFE evolve more smoothly than those of HMSE as the scale parameter increases. These results indicate that HMFE and HRCMFE capture the characteristics of white noise more effectively than HMSE.

Analyses with varying data lengths reveal that shorter data lengths degrade the computational stability of entropy values. For HMSE, the mean entropy curve demonstrates relatively smooth behavior at a data length of 4096; however, as the data length decreases, the curve exhibits more pronounced fluctuations, indicating that this method’s computational stability is significantly compromised by limited data. Notably, the standard deviation (SD) of HRCMFE remains smaller than that of HMFE and HMSE across all tested lengths, confirming that HRCMFE computation offers superior stability compared to the other two methods.

Collectively, these findings demonstrate that the proposed HRCMFE not only maintains robust computational stability but also effectively characterizes signal features, even under challenging data conditions.

## 5. Experiment Analysis

### 5.1. Experiments and Data Descriptions

To verify the proposed methods in fault diagnosis of planetary gearbox, a public failure dataset of a planetary gearbox named ‘WT-Planetary gearbox dataset’ was used in this study. The dataset was generated and collected by Liu et al. in a simulation test rig for a wind turbine planetary gearbox [13] and was publicly released in the paper ‘A review on deep learning in planetary gearbox health state recognition: methods, applications, and dataset publication’ to facilitate academic research and learning. An experimental platform was built to acquire the vibration data of a planetary gearbox under different operating states. The experimental platform included a motor, a planetary gearbox, a fixed-shaft gearbox, a load device, and a data collection device, as shown in Figure 5 [42]. The planetary gearbox contained one sun gear and four planet gears, and five different operation states were considered, as shown in Figure 6 [42]. Some of the important parameters are listed as follows: sun gear tooth number was 28, the ring gear tooth number was 100, the planet gear tooth number was 36, and the vibration data sampling frequency was 48 kHz. First, the feature extraction and fault diagnosis methods were validated under a constant speed condition (i.e., the sun gear rotational frequency was 20 Hz) in Section 5.2, Section 5.3 and Section 5.4. Subsequently, the proposed fault diagnosis framework was evaluated under different speed conditions.

The collected vibration data under different planetary gearbox states in the given time-domain are shown as Figure 7.

The dataset, publicly released by Liu et al., is large in volume: the collection duration exceeded 5 min [13]. A non-overlapping sample dataset was built to validate the effectiveness of the proposed approach. Initially, the original dataset corresponding to each operating state was divided into 200 groups, with 72,000 data points included in each group. Subsequently, samples featuring random starting positions were extracted from each group to construct a non-overlapping sample dataset, whose detailed parameters are presented in Table 2. For the LSSVM model, the total quantity of training samples amounted to 600, while the total number of testing samples reached 400.

### 5.2. Feature Extraction

HMSE, HMFE, and the proposed HRCMFE were used for feature extraction from the planetary gearbox sample dataset. The calculation parameters of the different methods were set the same as those in Table 1, except the number of hierarchies was set to 2. The mean and SD values of different entropies for 200 groups of vibration data under missing tooth states are shown in Figure 8.

As shown in Figure 8, for each hierarchical component, the mean entropy curves of HMFE and HRCMFE exhibit smoother transitions than those of HMSE as the multiscale entropy scale parameter increases. These observations indicate that HMFE and HRCMFE more effectively characterize the multiscale features of original signals compared to HMSE. Additionally, the standard deviation (SD) values of HRCMFE remain smaller than those of the other two methods across all tested scales. This result demonstrates that the proposed HRCMFE exhibits superior stability during entropy computation for vibration signals from planetary gearboxes.

To demonstrate the feature extraction effectiveness of HRCMFE under different operating states of planetary gearboxes, HRCMFE values were calculated for sample vibration datasets under five different operating states. The mean and SD values of HRCMFE across 200 groups of vibration data under the five operating states are shown in Figure 9.

As shown in Figure 9, the RCMFE values of vibration signals in different hierarchical components exhibit significant differentiation. The features of the root crack state can be clearly distinguished from those under other states in hierarchical component 1. The features of the broken tooth and healthy states exhibit overlapping phenomena in hierarchical components 1 and 3. However, these two states exhibit a clear distinction in hierarchical component 4. The results indicate that the characteristics of the two states display clear differences under high frequency conditions. The features of the wear gear state across all hierarchical components are relatively consistent, and the features of the low-scale RCMFE can be easily distinguished from those in other states. The features of the missing tooth state also exhibit a clear distinction from those in other states in the low-scale RCMFE of hierarchical components 2, 3, and 4.

Due to the fact that HRCMFE, HMFE, and HMSE extract features with multiple scales, intuitively assessing the influence of different features on classification via multi-dimensional comparisons is challenging. To address this, the trace ratio of the between-class scatter matrix to the within-class scatter matrix [50,51] is employed to quantify feature classifiability. A larger trace ratio signifies clearer inter-class sample separation, tighter intra-class sample aggregation, and superior class discriminability. The trace ratios of sample features extracted by the three methods are presented in Table 3.

As can be seen from Table 3, the feature extraction method based on HRCMFE has the largest trace ratio, demonstrating that its extracted features exhibit superior classifiability compared to those from other methods.

The results presented above indicate that feature extraction based on HRCMFE can effectively reflect the operating states of planetary gearboxes.

### 5.3. Fault Diagnosis by GWO-LSSVM

In this paper, GWO-LSSVM is proposed to improve the accuracy of fault diagnosis. The software of GWO can be obtained from the following link: http://www.mathworks.com.au/matlabcentral/fileexchange/44974-grey-wolf-optimizer--gwo- (accessed on 16 February 2025). The software of LSSVM can be obtained from the following link: http://www.esat.kuleuven.be/sista/lssvmlab (accessed on 8 April 2025). In order to verify the effectiveness of the proposed method, the fault diagnosis of the proposed method is compared to LSSVM. The parameters of the proposed GWO-LSSVM were set as follows: the optimization ranges of γ and σ were set to [0.01, 100], the wolf population size was set to 50, the maximum number of iterations was set to 30, and the number of subsets *M* was set as 10. The training feature datasets were extracted by the HRCMFE with the training samples described in Table 2. The fitness curve of parameter optimization by GWO was presented in Figure 10.

As shown in Figure 10, the diagnostic error rate of LSSVM was significantly reduced after optimization using GWO. Meanwhile, GWO demonstrates the ability to rapidly converge to the optimal solution within fewer iterations, which indicates its strong optimization performance.

The testing feature datasets extracted by HRCMFE using the testing samples described in Table 2 were applied to both GWO-LSSVM and LSSVM. The confusion matrix of the fault diagnosis results is shown in Figure 11.

As shown in Figure 11, LSSVM achieved 91% and GWO-LSSVM reached 99.2% in overall diagnostic accuracy. The proposed method demonstrates a significant improvement in fault diagnosis accuracy. The misdiagnosis of LSSVM mainly occurs between the healthy state and broken-tooth state, as well as between the missing-tooth state and gear wear state. GWO-LSSVM only misdiagnosed the missing-tooth state, while achieving 100% accuracy for fault diagnosis of all other states. The results indicate that the classification ability of LSSVM was enhanced by parameter optimization using GWO.

### 5.4. Comparison with Other Methods

To further evaluate the effectiveness of the proposed method in this paper, the other feature extraction methods, such as HMSE, HMFE, RCMSE, and RCMFE were applied for comparison. The software for RCMSE can be obtained from the following link: https://github.com/HamedAzami/NLDyn/blob/main/Final_V10p.rar. The parameters of HMSE and HMFE are listed in Table 1, and the parameters of RCMSE and RCMFE were set as follows: the scale factor was 20, the scalar embedding value was 2, the scalar time lag value was 1, and the scalar threshold value was 0.15. Backpropagation neural networks (BPNN), random forests (RF), least squares support vector machines (LSSVM), and gray wolf optimization-optimized LSSVM (GWO-LSSVM) were employed for fault diagnosis. The main parameters of LSSVM and GWO-LSSVM were defined in Section 5.3, while those of BPNN and RF are listed in Table 4. Fault diagnosis results from different feature extraction and classification methods are presented in Figure 12.

As shown in Figure 12, when using the same classification model, fault diagnosis accuracy with HRCMFE-based feature extraction surpasses that of other feature extraction methods. Refined composite entropy-based techniques, such as HRCMFE, demonstrate greater effectiveness than non-entropy-based approaches when paired with identical classification algorithms. For a given feature extraction method, GWO-LSSVM outperforms other classifiers in terms of classification performance. The proposed method achieves a diagnosis accuracy of 99.25%, the highest among all tested combinations. These results confirm that HRCMFE effectively captures the operational states of planetary gearboxes and that optimizing LSSVM parameters via GWO is critical for enhancing diagnostic accuracy.

### 5.5. Fault Diagnosis for Different Operation Conditons

In this section, vibration signal datasets collected under different sun gear rotation speeds are utilized to validate the fault diagnosis effectiveness of the proposed method. The structural organization of datasets for different operating conditions remains consistent with the framework outlined in Table 1. Fault diagnosis results are presented in Figure 13.

As shown in Figure 13, the proposed method in this paper maintains high diagnostic accuracy under different operating conditions of the planetary gearbox, demonstrating its robust generalization capability.

## 6. Conclusions

Efficient extraction and classification of fault features are critical challenges in planetary gearbox fault diagnosis. In this study, a novel feature extraction method termed HRCMFE and a classification method termed GWO-LSSVM were proposed to enhance the effectiveness of fault diagnosis for planetary gearboxes. The following conclusions were drawn from the simulations and experimental studies:(a)Simulation results demonstrate that HRCMFE exhibits better computational stability than HMFE and HMSE when processing white noise signals.(b)In comprehensive experimental validation, the HRCMFE-based fault feature extraction method for planetary gearboxes demonstrates superior effectiveness compared with methods based on HMSE, HMFE, RCMSE, and RCMFE.(c)The GWO-optimized LSSVM applied to planetary gearbox fault diagnosis shows statistically significant improvements in classification accuracy.(d)The method proposed in this paper exhibits good diagnostic efficiency under different work conditions. However, further investigation into its resilience under highly variable operational scenarios is warranted.

## Figures and Tables

**Figure 1 entropy-27-00512-f001:**
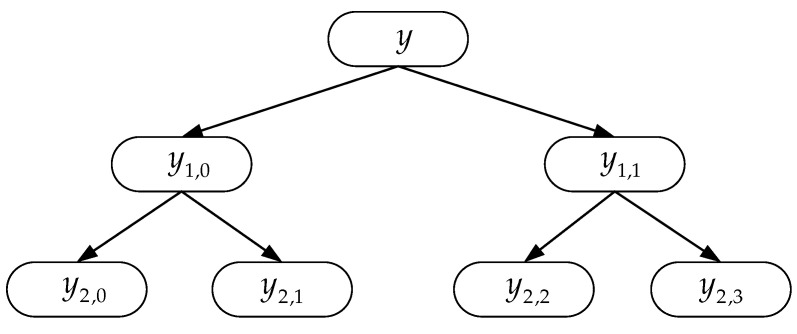
Schematic diagram of hierarchical segmentation of time Series (*h* = 2).

**Figure 2 entropy-27-00512-f002:**
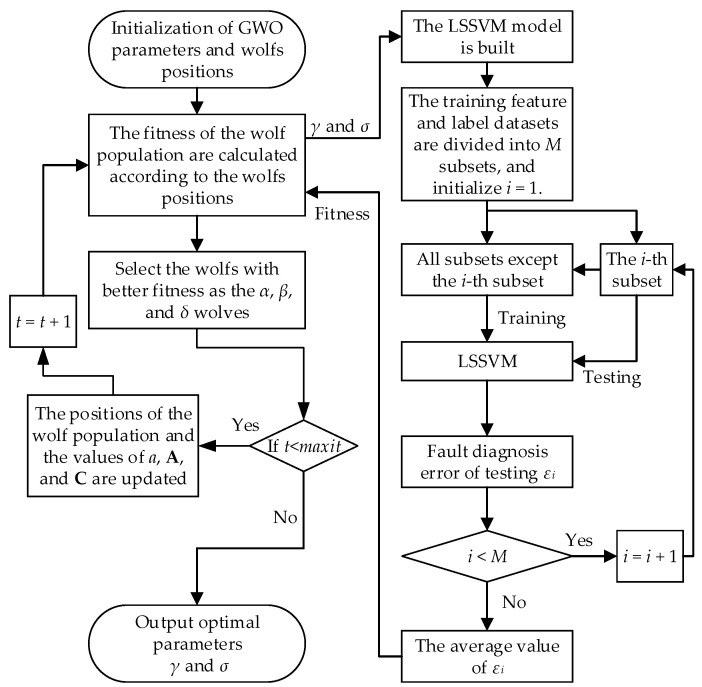
The flowchart of the proposed GWO-LSSVM.

**Figure 3 entropy-27-00512-f003:**
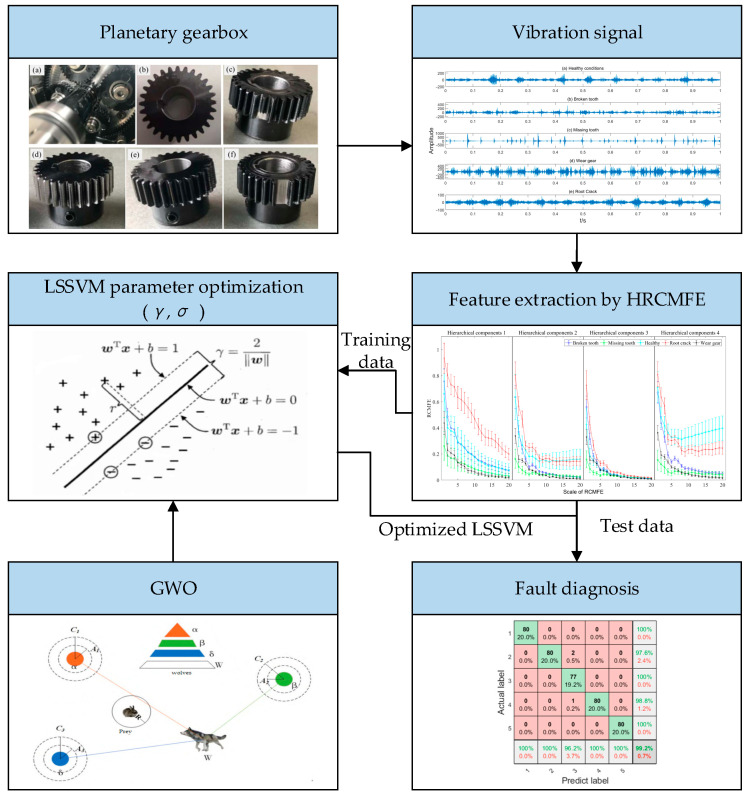
The framework of the proposed fault diagnosis method.

**Figure 4 entropy-27-00512-f004:**
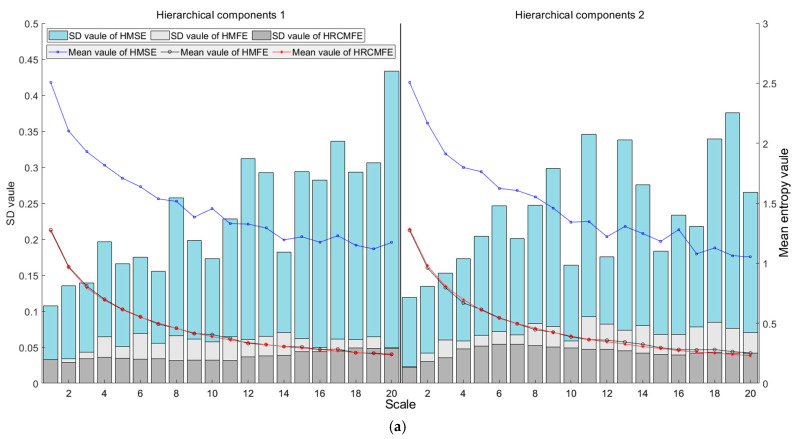
The mean value curve and SD value of HMSE, HMFE, and HRCMFE for white noise with different data lengths: (**a**) 1024 data length; (**b**) 2048 data length; (**c**) 4096 data length.

**Figure 5 entropy-27-00512-f005:**
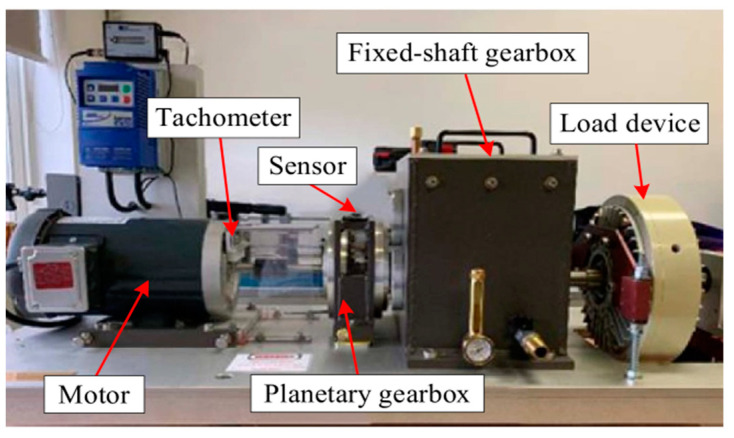
Diagram of experimental platform.

**Figure 6 entropy-27-00512-f006:**
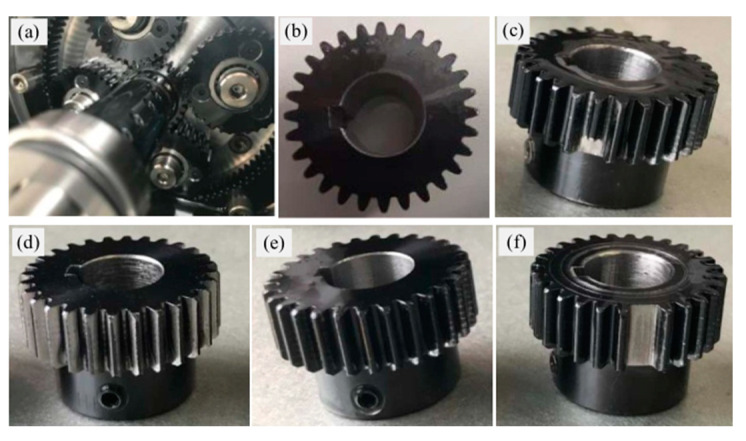
Planetary gearbox. (**a**) The structure of planetary gearbox, (**b**) healthy, (**c**) broken tooth, (**d**) wear gear, (**e**) root crack, (**f**) missing tooth.

**Figure 7 entropy-27-00512-f007:**
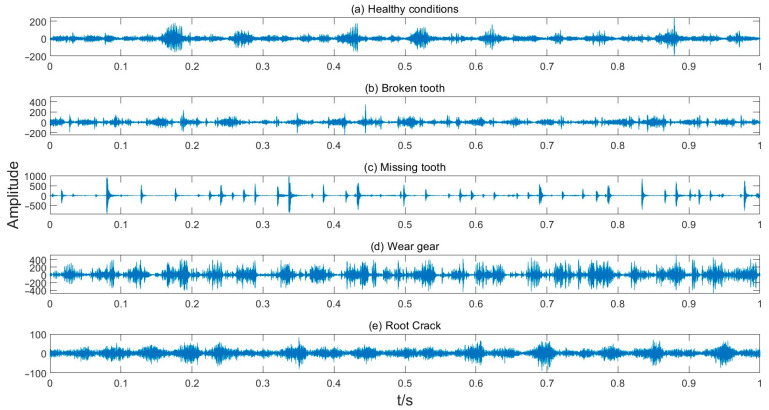
The time-domain vibration data of planetary gearbox.

**Figure 8 entropy-27-00512-f008:**
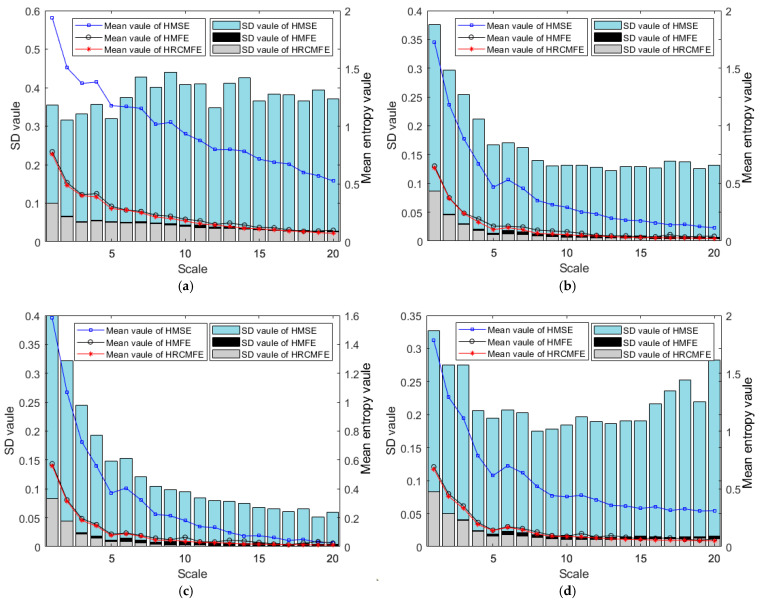
The mean value curve and SD value of HMSE, HMFE, and HRCMFE. (**a**) Hierarchical component 1, (**b**) hierarchical component 2, (**c**) hierarchical component 3, (**d**) hierarchical component 4.

**Figure 9 entropy-27-00512-f009:**
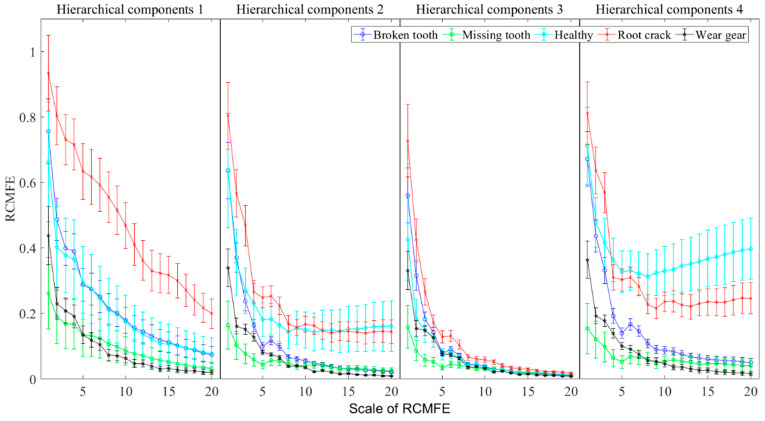
The mean value curve and SD value of HRCMFE under five operation states.

**Figure 10 entropy-27-00512-f010:**
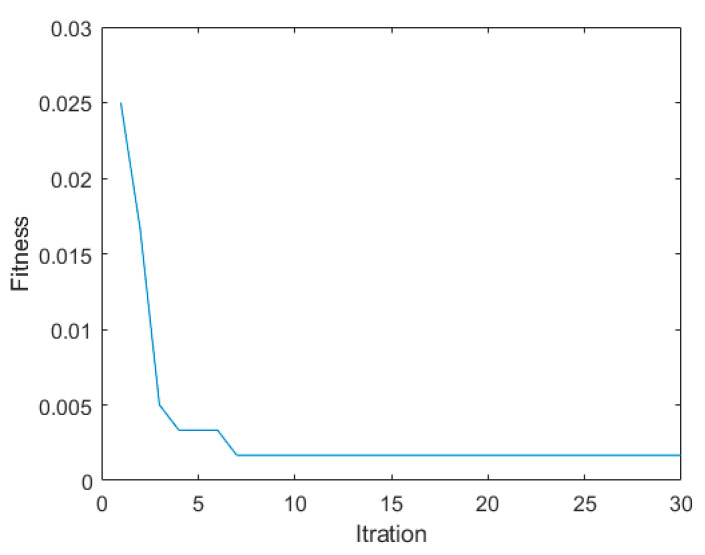
The fitness of GWO-LSSVM.

**Figure 11 entropy-27-00512-f011:**
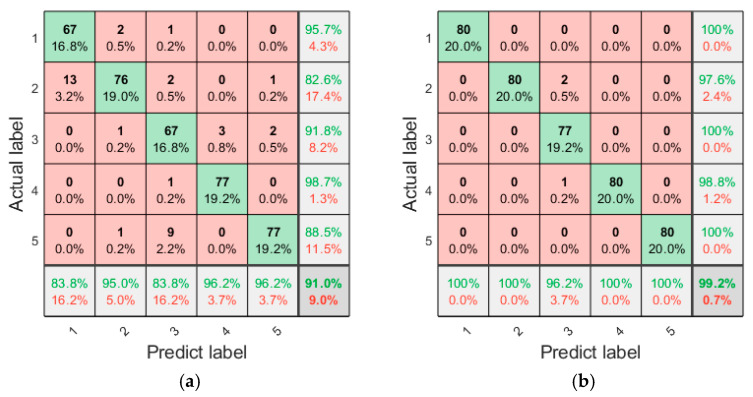
The confusion matrix of the fault diagnosis results (green represents correctly classified, and red represents incorrectly classified). (**a**) LSSVM, (**b**) GWO-LSSVM.

**Figure 12 entropy-27-00512-f012:**
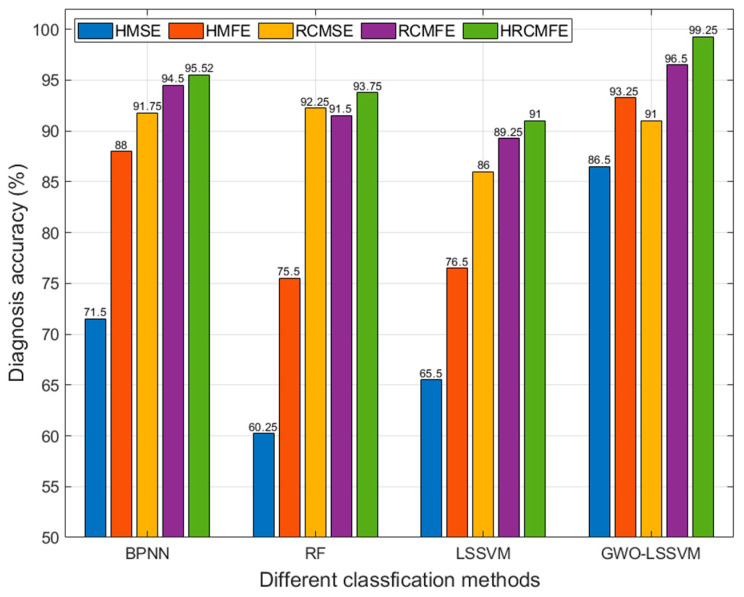
Comparison of fault diagnosis accuracy of different methods.

**Figure 13 entropy-27-00512-f013:**
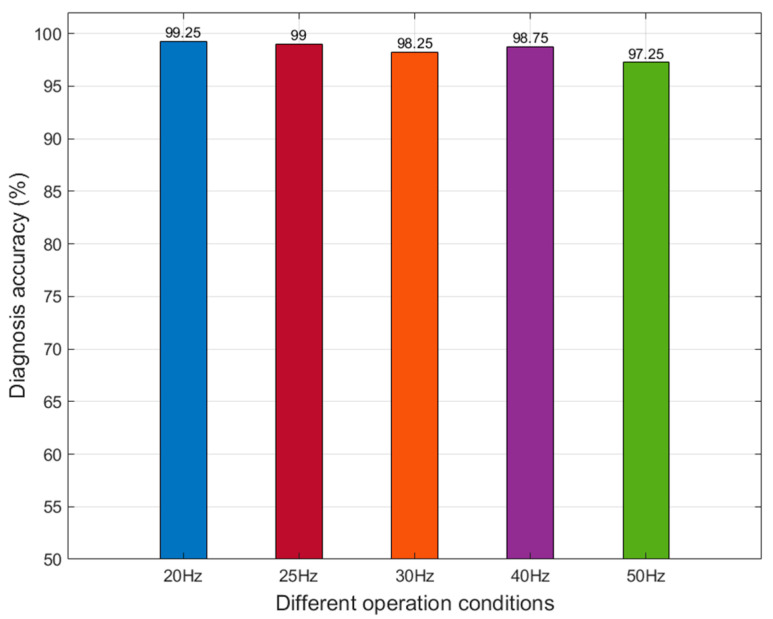
Comparison of fault diagnosis accuracy under different operation conditions.

**Table 1 entropy-27-00512-t001:** The parameter settings for the different methods.

Method	Special Parameters	Common Parameters
HMSE	/	Number of hierarchies is 1 Scale factor is 20 Scalar embedding value is 2 Scalar time lag value is 1 Scalar threshold value is 0.15
HMFE	Fuzzy power is 2	
HRCMFE	Fuzzy power is 2	

**Table 2 entropy-27-00512-t002:** Parameters of the sample dataset.

Operation State	Data Length	Number of Training Sample	Number of Testing Sample	Operation Label
Healthy	1024	120	80	1
Broken tooth		120	80	2
Missing tooth		120	80	3
Root crack		120	80	4
Wear gear		120	80	5

**Table 3 entropy-27-00512-t003:** Trace ratio of between-class scatter matrix and within class scatter matrix.

Feature Extraction Method	HRCMFE	HMFE	HMSE
Trace ratio	1.222	1.136	1.127

**Table 4 entropy-27-00512-t004:** The parameters of BPNN and RF.

Classification Methods	Parameters
BNPP	The number of hidden layer neuron is 10, learning rate is 0.01 maximum iterations is 1000, target training error is 10^−6^
RF	The number of trees is 100

## Data Availability

The data that support the findings of this study are available upon request from the authors.

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
