# Peer review of "Fault Diagnosis of Planetary Gearbox Based on Hierarchical Refined Composite Multiscale Fuzzy Entropy and Optimized LSSVM"

_entropy, 2025, doi:10.3390/e27050512_

Round 1
Reviewer 1 Report
Comments and Suggestions for Authors
This manuscript proposes a hybrid fault diagnosis framework for planetary gearboxes, addressing a significant problem in the field of rotating machinery diagnostics. Overall, the manuscript is well-organized, and the methodology is clearly described. However, several critical issues must be addressed before the manuscript can be considered for publication:
- The simulation study of the proposed HRCMFE is currently embedded within the methodology section. It is suggested that this simulation analysis be moved to the experimental section.
- Figures 2 and 8 present the results of HMSE, HMFE, and HRCMFE. However, the accompanying descriptions are difficult to follow. The use of a stacked bar chart organized across different methods creates confusion regarding the figure’s intent. It is unclear whether the purpose is to select the best scale or to compare the overall performance of each method.
- The improved performance of the proposed method comes from the combination of different parts. However, no analysis is provided regarding the individual contribution of each component. Thus, ablation studies should be conducted.
- Feature extraction in fault diagnosis can be considered a physics-knowledge-guided procedure. It would strengthen the manuscript to discuss potential connections between the proposed method and the concept of physics-guided modeling, which is presented in "Yin et al., Physics-guided degradation trajectory modeling for remaining useful life prediction of rolling bearings." It is recommended to include such a discussion in the introduction to enhance the paper's completeness.
- While the manuscript demonstrates that the proposed method outperforms traditional entropy methods and vanilla LSSVM, it does not include comparisons with other widely used machine learning algorithms, such as Random Forests and Backpropagation (BP) neural networks. A broader comparative analysis would significantly improve the strength of the results.
Author Response
Response to Reviewer Comments This manuscript proposes a hybrid fault diagnosis framework for planetary gearboxes, addressing a significant problem in the field of rotating machinery diagnostics. Overall, the manuscript is well-organized, and the methodology is clearly described. However, several critical issues must be addressed before the manuscript can be considered for publication: 1. The simulation study of the proposed HRCMFE is currently embedded within the methodology section. It is suggested that this simulation analysis be moved to the experimental section. Answer: Thanks for your suggestion. The simulation study had been moved to the Section 4 as the Simulation study part. 2. Figures 2 and 8 present the results of HMSE, HMFE, and HRCMFE. However, the accompanying descriptions are difficult to follow. The use of a stacked bar chart organized across different methods creates confusion regarding the figure’s intent. It is unclear whether the purpose is to select the best scale or to compare the overall performance of each method. Answer: Thank you for reminding us of this problem. Your question regarding the visual presentation and descriptive clarity of Figure 2 (currently Figure 4) and Figure 8 is critically important, providing significant guidance for enhancing the quality of the paper. We have revised the explanatory descriptions of these figures, emphasizing the use of figure comparisons to illustrate the computational stability of the proposed method. 3. The improved performance of the proposed method comes from the combination of different parts. However, no analysis is provided regarding the individual contribution of each component. Thus, ablation studies should be conducted. Answer: Thanks for your suggestion. We have added additional content to elaborate on the roles of each component in the proposed method. Specifically, the trace ratio is employed to analyze the features extracted by different feature extraction methods, aiming to validate the classifiability of the HRCMFE method proposed in this paper. Furthermore, classification performance comparisons with other methods are conducted to verify the classification capability of the GWO-LSSVM approach presented herein. 4. Feature extraction in fault diagnosis can be considered a physics-knowledge-guided procedure. It would strengthen the manuscript to discuss potential connections between the proposed method and the concept of physics-guided modeling, which is presented in "Yin et al., Physics-guided degradation trajectory modeling for remaining useful life prediction of rolling bearings." It is recommended to include such a discussion in the introduction to enhance the paper's completeness. Answer: We appreciate your suggestion. We have added a comprehensive review of current physics-based signal analysis methods in the Introduction section, strengthened the completeness of the literature review, and reference relevant papers. 5. While the manuscript demonstrates that the proposed method outperforms traditional entropy methods and vanilla LSSVM, it does not include comparisons with other widely used machine learning algorithms, such as Random Forests and Backpropagation (BP) neural networks. A broader comparative analysis would significantly improve the strength of the results. Answer: Thanks for your suggestion. In Section 5.4, we have included the commonly used BPNN (Backpropagation Neural Network) and RF (Random Forest) classification methods for comparative analysis to validate the effectiveness of the proposed classification approach.
Reviewer 2 Report
Comments and Suggestions for Authors
This paper proposes a method based on hierarchical refined composite multiscale fuzzy entropy and optimized LSSVM for planetary gearbox fault diagnosis. This paper is well organized and the proposed method is described and validated. However, the following deficiencies need to be carefully addressed to further improve the quality of the paper.
- The literature review is not comprehensive and lacks a brief comparison of rotating machinery fault diagnosis methods based on signal processing technology. For example: 10.1016/j.ymssp.2012.06.021, 10.1088/1361-6501/ab02d8, 10.1016/j.ymssp.2024.112262,10.1016/j.ymssp.2024.112026.
- Currently, deep learning based fault diagnosis methods are popular. A comparison with deep learning-based methods needs to be briefly discussed in Section 1.
- In Sections 2 and 3, important formulas need to provide corresponding references.
- Symbols representing vectors and matrices should be bolded to distinguish them, such as u and y. Avoid using the same symbol to express different meanings, such as D. Check the symbols in the manuscript carefully.
- Some formulas have problems. For example, it is unclear how MFE equals FuzzyEn in Eq. (6); h in Eq. (10) seems to be k; Lssvm_model is not used in Eq. (23).
- Eliminate grammatical errors, such as lines 158 and 159; the two paragraphs below Figure 9 are repeated. The manuscript needs to be proofread carefully.
- Only one experimental case verification is insufficient, and more experimental case verifications need to be supplemented. The verification results under different operating conditions are not shown, which makes it difficult to illustrate the generalization ability of the proposed method.
- Lack of comparison with other classification methods.
- Avoid reintroducing the manuscript methodology in Section 5. Section 5 should focuses on the main findings and conclusions. The limitations of the proposed method should be discussed, such as under variable operating conditions.
Author Response
Response to Reviewer Comments
This paper proposes a method based on hierarchical refined composite multiscale fuzzy entropy and optimized LSSVM for planetary gearbox fault diagnosis. This paper is well organized and the proposed method is described and validated. However, the following deficiencies need to be carefully addressed to further improve the quality of the paper.
1.The literature review is not comprehensive and lacks a brief comparison of rotating machinery fault diagnosis methods based on signal processing technology. For example: 10.1016/j.ymssp.2012.06.021, 10.1088/1361-6501/ab02d8, 10.1016/j.ymssp.2024.112262,10.1016/j.ymssp.2024.112026.
Answer: Thanks for your suggestion. We have added a comprehensive review of current physics-based signal analysis methods in the Introduction section, strengthened the completeness of the literature review, and reference relevant papers.
2. Currently, deep learning based fault diagnosis methods are popular. A comparison with deep learning-based methods needs to be briefly discussed in Section 1.
Answer: Thank you for reminding us of this problem. We have added a discussion on deep learning-based fault diagnosis methods in Section 1.
3. In Sections 2 and 3, important formulas need to provide corresponding references.
Answer: Thank you for reminding us of this problem. Following your recommendations, we have added relevant references for several key formulas and theories discussed in the paper.
4. Symbols representing vectors and matrices should be bolded to distinguish them, such as u and y. Avoid using the same symbol to express different meanings, such as D. Check the symbols in the manuscript carefully.
Answer: Thanks for your suggestion. We have checked and corrected such errors. However, since some of these are time series, which we do not consider vectors, they were not bolded.
5. Some formulas have problems. For example, it is unclear how MFE equals FuzzyEn in Eq. (6); h in Eq. (10) seems to be k; Lssvm_model is not used in Eq. (23).
Answer: Thank you for reminding us of this problem. We have provided an explanation of the formula below Eq. (6), clarifying its calculation process. Some errors in the parameter settings have been identified and corrected, specifically in Eq. (10), Eq. (11) and Eq. (23).
6. Eliminate grammatical errors, such as lines 158 and 159; the two paragraphs below Figure 9 are repeated. The manuscript needs to be proofread carefully.
Answer: Thank you for reminding us of this problem. We have carefully corrected the grammatical errors and spelling mistakes in the manuscript, and also addressed the issues of repetitive content.
7. Only one experimental case verification is insufficient, and more experimental case verifications need to be supplemented. The verification results under different operating conditions are not shown, which makes it difficult to illustrate the generalization ability of the proposed method.
Answer: Thanks for your suggestion. In Section 5.5, we have added fault diagnosis analysis for planetary gearboxes under different rotational speed conditions to validate the generalization ability of the proposed method.
8. Lack of comparison with other classification methods.
Answer: Thanks for your suggestion. In Section 5.4, we have included the commonly used BPNN (Backpropagation Neural Network) and RF (Random Forest) classification methods for comparative analysis to validate the effectiveness of the proposed classification approach.
9. Avoid reintroducing the manuscript methodology in Section 5. Section 5 should focuses on the main findings and conclusions. The limitations of the proposed method should be discussed, such as under variable operating conditions.
Answer: Thanks for your suggestion. We have rewritten the conclusion section and provided a detailed elaboration on the study's limitations as well as our future research plans.

Round 2
Reviewer 2 Report
Comments and Suggestions for Authors
The authors have made appropriate revisions and additions to enhance the quanlity of the manuscript. I have no further questions.
Author Response
Comments and Suggestions for Authors
The authors have made appropriate revisions and additions to enhance the quanlity of the manuscript. I have no further questions.
Answer:There are no questions from the reviewer.